# Health itinerary-related survival of children under-five with severe malaria or bloodstream infection, DR Congo

Bieke Tack[1,2,3]*, Daniel Vita[4], José Nketo[5], Naomie Wasolua[4], Nathalie Ndengila[4], Natacha Herssens[1], Emmanuel Ntangu[4], Grace Kasidiko[4], Gaëlle Nkoji-Tunda[6,7], Marie-France Phoba[6,7], Justin Im[8], Hyon Jin Jeon[8,9], Florian Marks[8,9,10,11], Jaan Toelen[3,12], Octavie Lunguya[6,7], Jan Jacobs[1,2]

1 Department of Clinical Sciences, Institute of Tropical Medicine, Antwerp, Belgium, 2 Department of Microbiology, Immunology and Transplantation, KU Leuven, Leuven, Belgium, 3 Department of Pediatrics, University Hospitals UZ Leuven, Leuven, Belgium, 4 Hôpital Général de Référence Saint Luc de Kisantu, Kisantu, Democratic Republic of the Congo, 5 Zone de Santé Kisantu, Kisantu, Democratic Republic of the Congo, 6 Department of Microbiology, Institut National de Recherche Biomédicale, Kinshasa, Democratic Republic of the Congo, 7 Department of Medical Biology, University Teaching Hospital of Kinshasa, Kinshasa, Democratic Republic of the Congo, 8 International Vaccine Institute, Seoul, Republic of Korea, 9 Cambridge Institute of Therapeutic Immunology and Infectious Disease, University of Cambridge School of Clinical Medicine, Cambridge, United Kingdom, 10 Heidelberg Institute of Global Health, University of Heidelberg, Heidelberg, Germany, 11 Madagascar Institute for Vaccine Research, University of Antananarivo, Antananarivo, Madagascar, 12 Department of Development and Regeneration, KU Leuven, Leuven, Belgium

* btack@itg.be

## Abstract

### Background

Prompt appropriate treatment reduces mortality of severe febrile illness in sub-Saharan Africa. We studied the health itinerary of children under-five admitted to the hospital with severe febrile illness in a setting endemic for *Plasmodium falciparum* (*Pf*) malaria and invasive non-typhoidal *Salmonella* infections, identified delaying factors and assessed their associations with in-hospital death.

### Methodology

Health itinerary data of this cohort study were collected during 6 months by interviewing caretakers of children (>28 days − <5 years) admitted with suspected bloodstream infection to Kisantu district hospital, DR Congo. The cohort was followed until discharge to assess in-hospital death.

### Principal findings

From 784 enrolled children, 36.1% were admitted >3 days after fever onset. This long health itinerary was more frequent in children with bacterial bloodstream infection (52.9% (63/119)) than in children with severe *Pf* malaria (31.0% (97/313)). Long health itinerary was associated with in-hospital death (OR = 2.1, p = 0.007) and two thirds of deaths occurred during the first 3 days of admission. Case fatality was higher in bloodstream infection (22.8% (26/

**Data Availability Statement:** Due to the sensitive nature of the data, the authors are unable to share the data directly. Requests to access the data can be made to ITM's contact point for data access

(ITMresearchdataaccess@itg.be). All requests will be reviewed for approval by ITMs Data Access Committee. After approval, data sharing will be managed by the same committee.

**Funding:** This research was funded by the Belgian Directorate of Development Cooperation and Humanitarian Aid (DGD) through Framework Agreement between the Belgian DGD and the Institute of Tropical Medicine, Belgium. B.T. has a scholarship from Research Foundation Flanders (FWO, 1153220N & 1153222N) and received a research grant for the follow-up study 'Treating Non-typhoidal Salmonella Bloodstream Infections in Children Under-five in DR Congo: a Cohort Study (TreNTS)' from the European Society of Clinical Microbiology and Infectious Diseases (ESCMID) in 2020. Part of the blood culture surveillance activities were supported by the Bill and Melinda Gates Foundation project OPP1127988 funded to International Vaccine Institute. This study has also received funding from the European Union's Horizon 2020 research and innovation program under the Vacc-iNTS project, grant agreement No 815439. The funders had no role in study design, data collection and analysis, decision to publish, or preparation of the manuscript.

**Competing interests:** The authors declare no conflict of interest.

114)) compared to severe *Pf* malaria (2.6%, 8/309). Bloodstream infections were mainly (74.8% (89/119)) caused by non-typhoidal *Salmonella*. Bloodstream infections occurred in 20/43 children who died in-hospital before possible enrolment and non-typhoidal *Salmonell*a caused 16 out of these 20 bloodstream infections. Delaying factors associated with in-hospital death were consulting traditional, private and/or multiple providers, rural residence, pre-hospital intravenous therapy, and prehospital overnight stays. Use of antibiotics reserved for hospital use, intravenous therapy and prehospital overnight stays were most frequent in the private sector.

## Conclusions

Long health itineraries delayed appropriate treatment of bloodstream infections in children under-five and were associated with increased in-hospital mortality. Non-typhoidal *Salmonella* were the main cause of bloodstream infection and had high case fatality.

## Trial registration

NCT04289688

### Author summary

Severe febrile illnesses, including severe malaria and bacterial bloodstream infections, are a major cause of under-five mortality. To save lives, on-time hospital referral is crucial to treat severe febrile illness appropriately as soon as possible. In a district hospital in DR Congo, we observed that one third of children with severe febrile illness had a long health itinerary, which meant that they arrived at the hospital >3 days after fever onset. Long health itinerary was most frequent (50.8%) in children with bloodstream infection. In-hospital death was more frequent in children with bloodstream infection (22.8%) than in children with severe malaria (2.6%). In-hospital death mostly occurred early after hospital arrival (admission day 1–2) and having a long health itinerary was associated with death. Finally, we observed that consulting multiple prehospital healthcare providers and prehospital injections & overnight stays were associated with long health itinerary and in-hospital death. We conclude that children with severe febrile illness often arrived late at the hospital, which might contribute to high case fatality, particularly for children with bloodstream infection. Future research and interventions should focus on improved danger sign recognition and earlier referral of children with bloodstream infections by (in)formal frontline healthcare workers.

## Introduction

In sub-Saharan Africa, severe febrile illness accounts for most of the under-five mortality [1,2] and is a major cause of hospital admission [3]. Main causes of severe febrile illness are *Plasmodium falciparum* (*Pf*) malaria, pneumonia, and bacterial bloodstream infections [2–5]. Due to the limited availability of blood culture diagnostics, community-acquired bloodstream infections are frequently overlooked and misdiagnosed as severe *Pf* malaria [3–5]. Bloodstream infections are detected in approximately 10% of hospital-admitted children with severe febrile illness and their case fatalities exceed 15% [4,5]. Non-typhoidal

*Salmonella* are a predominant cause of bloodstream infection, particularly in children with (recent) *Pf* malaria infection [4–6].

Starting appropriate treatment of severe febrile illness within 24 to 48 hours is crucial to avoid progression to life-threatening sepsis and requires hospital admission [7–9] and thus reduces mortality [8,9]. Unfortunately, in low-resource settings, delayed and inappropriate healthcare seeking and prehospital patient management, together with transport difficulties, delay appropriate treatment and are associated with in-hospital death [8–11].

Current knowledge of delaying factors is mostly derived from research on paediatric febrile illness in general or on childhood malaria, pneumonia, or diarrhoea [10]. For children with severe febrile illness and children with bloodstream infections in particular, data are scarce. It is however in this population that appropriate treatment is most urgent and requires early referral, as is stressed by the high mortality during the first 48 hours of sepsis treatment [5,8,12].

In this hospital-based study, we assessed the health itinerary of children under-five with severe febrile illness. Delaying factors were identified according to the Three Delays framework, which describes delays related to healthcare seeking, reaching the healthcare provider and prehospital appropriate patient management [8,11]. Next, we aimed to identify delaying factors associated with in-hospital death for prioritization during health education and health system reforms.

## Methods

### Ethics statement

The study was conducted in accordance with the principles of the Declaration of Helsinki and international scientific standards [13]. Written informed consent was given by the caretaker of each enrolled child. Ethical approval was granted by the Institutional Review Board of ITM (1367/20), the Ethics Committee of Antwerp University (20/14/172), and the Ecole de Santé Public Kinshasa (145/2020). Patients or the public were not involved in our research.

### Study design, period, population and setting

This prospective observational cohort study (Clinicaltrials.gov: NCT04289688, clinicaltrials. gov/ct2/show/NCT04289688) was organized at Hôpital St. Luc Kisantu (Kisantu hospital) from February to July 2021. All children aged >28 days and <5 years admitted to Kisantu hospital with severe febrile illness who fulfilled the criteria for blood culture sampling (criteria in S1 Table) were eligible. Data from children who had been previously enrolled in the study were excluded from the analysis.

Kisantu hospital is located 120km south of Kinshasa in Kongo Central Province and functions as referral hospital for 214,780 inhabitants (estimated population 2021) of Kisantu health district [14]. The district has a surface of approximately 1400km$^2$ and is subdivided in 17 health areas (small, neighbouring areas were aggregated in Fig 1), each served by minimum one registered health centre (Fig 1). In total, there are 50 publicly registered community healthcare facilities (27 health centres, 23 health posts). Registered health centres charge a flat fee (6500 CDF = 3.25 USD) per consultation, which includes basic diagnostic and therapeutic management. If patients are referred to Kisantu hospital by an official health centre, a flat fee (30,000 CDF = 15 USD) is charged for consultation and admission [15]. This fee covers their admission as well as basic diagnostic and therapeutic management for 14 days. Kisantu hospital is a 340-bed hospital with an average bed occupancy of 140% in the 84-bed paediatric ward.

In Kongo Central Province, *Pf* malaria transmission is high, stable, and year-round with an increase in the rainy season [14]. One in four children (6–59 months) had a positive blood

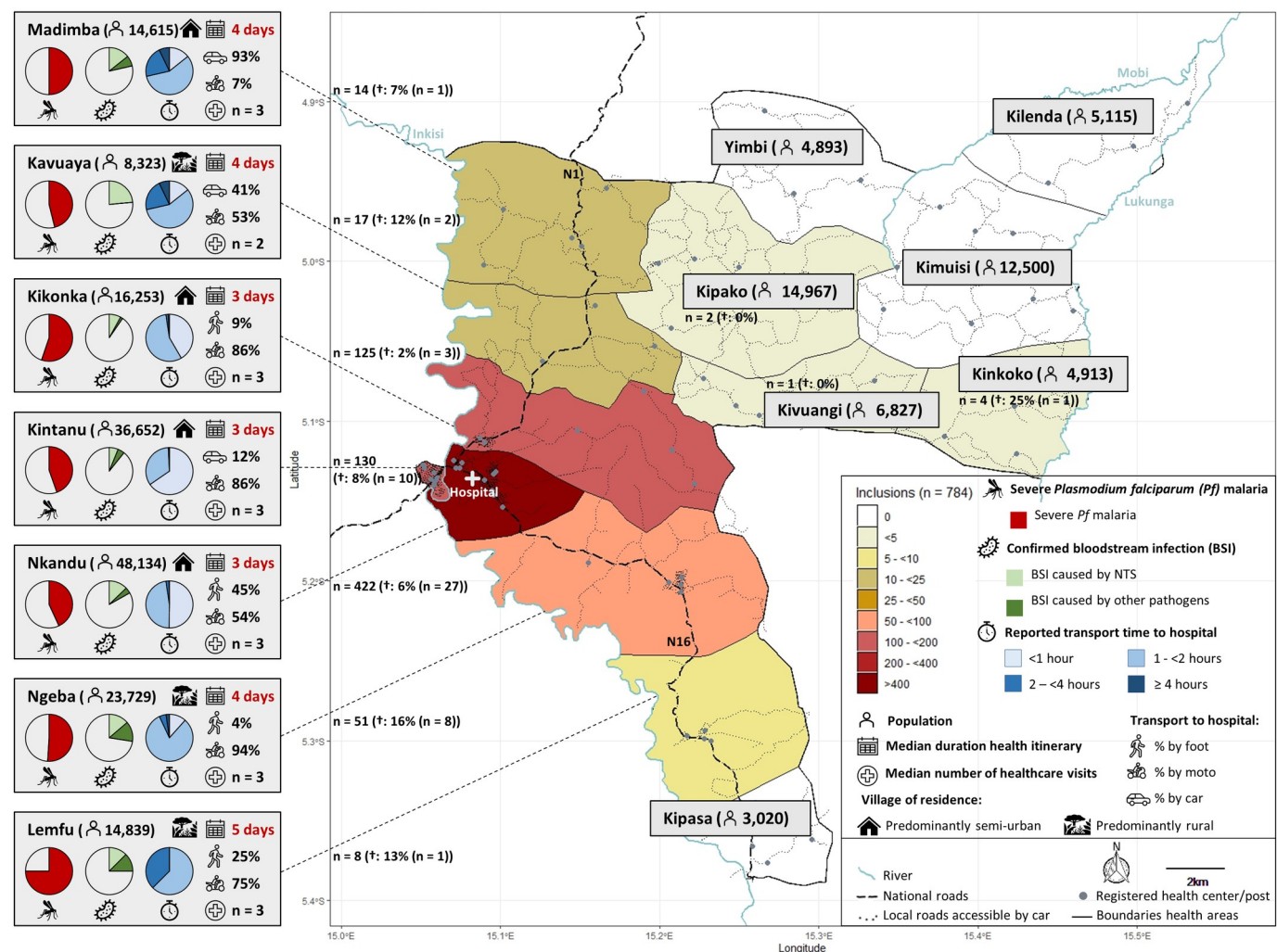

**Fig 1.** Geographical overview displaying (i) the population and healthcare services, (ii) the number of enrolled children and proportion of in-hospital deaths (†), (iii) the health itinerary (duration (time since fever onset), transport to hospital, number of visits) and (iv) the diagnostic category of enrolled children for each health area from Kisantu health district. The map was created with R-packages 'sf' & 'ggsn' and map base layer data are available open access from http://www.rgc.cd/. [15]Abbreviations: BSI: bloodstream infection; NTS: non-typhoidal *Salmonella; Pf: Plasmodium falciparum.*

microscopy during a national health survey in 2013–2014 [16]. This survey also demonstrated that 69% of children (6–59 months) were anaemic, that 11% of children (6–59 months) suffered from acute malnutrition and that HIV-prevalence was relatively low (0.2% HIV positivity in adult population) [16]. Although SARS-CoV-2 seroprevalence was high (>75%) in Kongo-Central Province in 2021, the number of reported or hospitalized cases was relatively low [17].

Since 2007, Kisantu hospital is part of a blood culture surveillance network organized by Institut National de Recherche Biomédicale (INRB; Kinshasa, DR Congo) and Institute of Tropical Medicine (ITM; Antwerp, Belgium) [18–22]. A blood culture (1–4 ml blood in BacT/ALERT bottle (bioMérieux, Marcy-l'Etoile, France)) is routinely sampled free-of-charge upon admission in all children suspected of having a bloodstream infection (S1 Table) [18,19]. Blood culture surveillance has revealed a high burden of non-typhoidal *Salmonella* (NTS) bloodstream infections with NTS causing three out of four confirmed bloodstream infections in children under-five admitted to Kisantu hospital [19].

### Eligibility screening, enrolment, and data collection

All children who presented at the paediatric ward were screened for eligibility by a trained study nurse. Reasons for non-eligibility were recorded (Fig 2). Caretakers of eligible children were asked for informed consent before enrolment and data collection. Enrolment and data collection occurred 7/7 days during working hours (8 am– 4 pm on weekdays, 8–12 am in weekend). Children who presented after working hours were enrolled the next morning.

Health itinerary refers to everything that happened with the child between the fever onset and hospital admission (Box 1). The health itinerary of each child was questioned via a semi-structured interview with the caretaker by a trained research nurse or physician as soon as possible after arrival of the child at the hospital. In addition, data on referral of the child were collected from the referral letter. All data were entered directly in an electronic case report form (RedCap, Vanderbilt University, Tennessee, US) on a tablet. For a pdf version of the electronic case report form we refer to S1 Document.

Venous blood for malaria microscopy examination and a blood culture were sampled and worked-up as part of routine patient care and blood culture surveillance. Blood culture work-up and pathogen identification occurred on site, according to previously published procedures [18–22]. Rapid malaria antigen tests (SD BIOLINE Malaria Ag P.f./Pan test 05FK60, Abbott, Chicago, US) were inoculated with capillary blood for each enrolled child. Together with the results from malaria microscopy, these tests allow to differentiate current (positive *Pf* malaria microscopy), recent (negative *Pf* malaria microscopy, *Pf*-HRP2 antigen positive and pan-LDH antigen negative) and very recent (negative *Pf* malaria microscopy, *Pf*-HRP2 antigen positive and pan-LDH antigen positive) *Pf* malaria infections [23,24]. Severe *Pf* malaria was defined according to World Health Organization (WHO) criteria [7]. NTS–*Pf* co-infection was defined as the copresence of a blood culture confirmed NTS bloodstream infection and a current or (very) recent *Pf* malaria infection [24]. Non-*Pf* malaria infections were microscopically confirmed malaria infections with species other than *P. falciparum*. *Enterococcus faecium*, s, *Staphylococcus aureus*, *Klebsiella pneumoniae*, *Acinetobacter baumannii*, *Pseudomonas aeruginosa*, *Enterobacter* spp. and *Escherichia* coli are together referred to as ESKAPE-E pathogens and are frequent causes of multidrug resistant hospital-associated infections [25]. Antibiotics were grouped according the WHO Access, Watch or Reserve (AWaRe) classification and their inclusion in the national essential medicines list [26,27]. Data on in-hospital outcome were collected by daily consultation of the medical files of enrolled children until discharge or death.

### Data analysis

Data analysis was performed in R version 3.6.1. Health itinerary was described according to the three-delay model by calculation of proportions and medians and their 25th and 75th percentiles (P25 –P75). Proportions were compared by calculation of the odds ratio and with a chi-squared test. Medians were compared with unpaired Wilcoxon rank-sum test. The map was created with R-packages 'sf' & 'ggsn', map base layer data are available open access from http://www.rgc.cd/ [15]. The Sankey diagram was created with R-package 'plotly'. In-hospital survival per diagnostic subgroup was compared by Kaplan-Meier survival analysis (R-packages 'survival' & 'survminer'). Unadjusted associations with long health itinerary and in-hospital death were assessed with logistic regression. Adjusted odds ratios were calculated by multivariable logistic regression per category of delay on a subset of variables with significant unadjusted associations with long health itinerary and in-hospital death respectively to avoid multicollinearity. The manuscript was written according to the STROBE guidelines for cohort reporting (S2 Table) [28].

**Children screened: n = 1597**

➤ *Not eligible due to age (≤28 days or ≥5 years): 467*
➤ *Not eligible due to no (history of) fever: 109*
➤ *Not eligible due to no suspicion of bloodstream infection: 31*
➤ *Not eligible due to death at arrival at the hospital: 8*

**Eligible children: n = 982 (61.5% of screened)**

➤ *Not enrolled due to death before enrolment: 43 (incl. 16 with NTS BSI, 4 with other BSI)*
➤ *Not enrolled due to child not retrieved: 22*
➤ *Not enrolled due to ambulatory care: 63*
➤ *Not enrolled due to admission to a non-pediatric ward: 18*
➤ *Not enrolled due to refusal to participate: 3*

**Enrolled children: n = 838 (85.3% of eligible)**

➤ *Children previously enrolled in the study: 54*

**Children enrolled for the 1st time: n = 784**

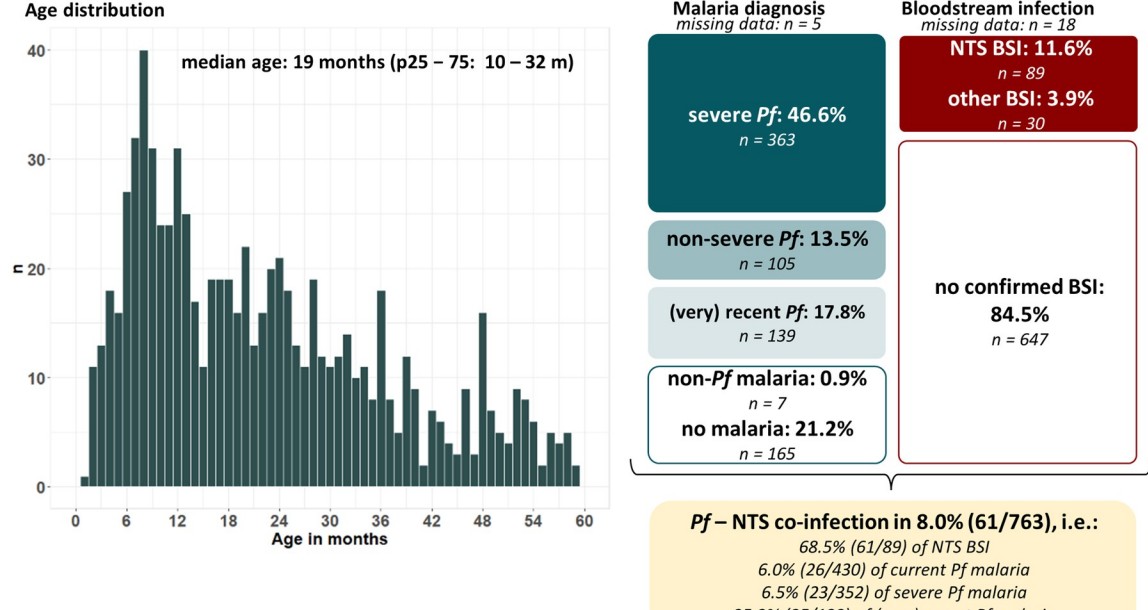

**Fig 2. Flowchart of study enrolment and overview of sociodemographic and diagnostic characteristics of study population.**
Abbreviations: BSI: bloodstream infection; NTS: non-typhoidal *Salmonella*; *Pf*: *Plasmodium falciparum*. *An admission in the last year was defined as minimum one overnight stay in a registered or private health centre or a hospital, preceding the fever onset of the current disease episode.

> ## Box 1. Definitions used to describe health itinerary
>
> **Health itinerary**: sequence of all actions of healthcare seeking and care provision (at home or from an external healthcare provider) between fever onset and hospital admission.
>
> **Long health itinerary:** hospital admission > 3 days after fever onset [64]
>
> **Home care:** administration of medicines that were still available at home, either remaining from prior illness or bought and stored at home in advance to illness [10]
>
> **Healthcare seeking:** any care sought from an external healthcare provider (briefly referred to as provider) [9]
>
> **Delayed first healthcare seeking:** first healthcare seeking > 1 day after fever onset [7,8,41,65]
>
> **Formal healthcare seeking:** healthcare seeking at registered health posts, health centres or hospital. These providers were grouped as formal healthcare providers [9,10,36,37].
>
> ➤ There are no other hospitals in Kisantu health district.
>
> ➤ **Health centres** are the main pillar of curative and preventive community healthcare services, e.g. curative care according to WHO IMCI recommendations or childhood immunization program. Remote or large health centres are often run by medical doctors and nurses, others by nurses only [15,56,57]
>
> ➤ **Health posts** offer a limited package of curative and preventive healthcare services compared to health centres and are run by nurses or health workers [14,15]
>
> **Informal healthcare seeking:** healthcare seeking from private pharmacists or drug vendors (further referred to as pharmacy), traditional healthcare providers (religious practitioners/institutions also considered as traditional), private practitioners or private health centres. These providers were considered as informal healthcare providers [9,10,36,37].
>
> **Delayed formal healthcare seeking:** formal healthcare seeking > 1 day after fever onset [7]
>
> **Caretaker:** person who sought care with the child at the hospital [9]
>
> **Overnight stay:** Passing the night in a healthcare facility before hospital admission

## Results

### Study population

During the six-month study period, 784 children were enrolled as "first-time" visit. The majority (60.3%, 473/784) were <2 years old. Fig 2 provides a detailed sociodemographic profile of the study population. Most (86.4%, 677/784) enrolled children were living in semi-urban health areas (Nkandu, Kintanu and Kikonka) surrounding the hospital, which together account for 47% of the population from Kisantu health district (Fig 1). Ten enrolled children were living outside of Kisantu health district.

Blood cultures were sampled and worked up for 97.7% (766/784) of children. A total of 122 bacterial pathogens were isolated in blood cultures of 15.5% (119/784) children. Non-typhoidal *Salmonella* (NTS) accounted for 74.8% (89/119) of bloodstream infections. Other

pathogens were *Staphylococcus aureus* (n = 11), *Klebsiella* spp. (n = 7, including 3 co-infections with NTS), *Escherichia coli* (n = 4), *Enterococcus* spp. (n = 3), *Salmonella* Typhi (n = 2), *Enterobacter* spp. (n = 2), *Serratia* spp. (n = 1), *Pseudomonas* spp. (n = 1) and *Acinetobacter* spp. (n = 1).

Complete results of malaria microscopy and rapid diagnostic test were available for 99.4% (779/784) of children. For five children microscopical *Plasmodium* species identification was missing but all were considered as *Pf* malaria based on *Pf*-HRP2 antigen positivity on rapid diagnostic test. Current malaria infection was diagnosed in 60.6% (475/784) and almost exclusively caused by *Pf* (Fig 2). Other *Plasmodium* species were *P. ovale* (n = 4), *P. malariae* (n = 3) and *P. falciparum–malariae* mixed infection (n = 2). Based on the result of the malaria rapid diagnostic test, almost half (45.7%, 139/304) of children with a negative malaria microscopy test had recent (n = 120) or very recent (n = 19) *Pf* malaria infection. A NTS–*Pf* co-infection was diagnosed in 8.0% (61/763) of children (Fig 2) and accounted for 6.0% of children with current *Pf* malaria and 51.3% of children with bloodstream infection.

## Health itinerary according to the three-delay model

**Long duration of health itinerary.** The median duration of health itinerary was 3 days (P25 – P75: 2–4 days) and 36.1% (283/784) of children had a long health itinerary duration (Fig 3). A long health itinerary was more prevalent in children with blood culture confirmed

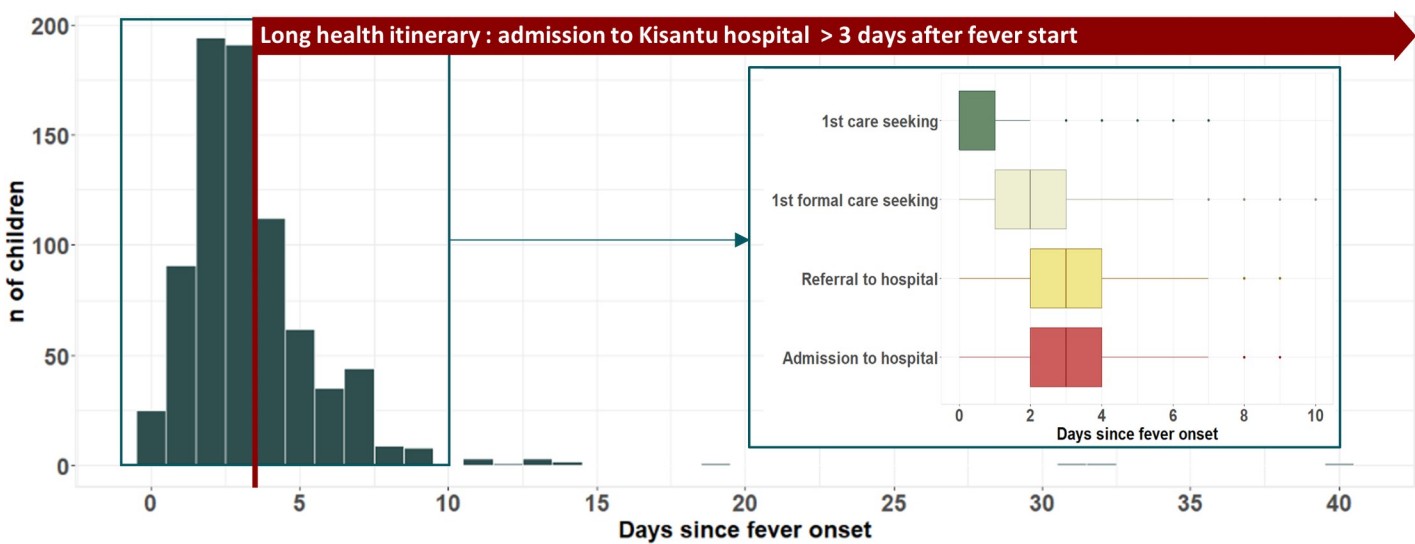

| *Health itinerary duration* | Overall<br>n = 784 | Severe<br>*Pf* malaria<br>n = 313 | NTS – *Pf*<br>co-infection<br>n = 61 | NTS BSI<br>n = 28 | Other BSI<br>(± *Pf* malaria)<br>n = 30 | Other<br>febrile illness<br>n = 331 |
|---|---|---|---|---|---|---|
| Median (P25 – P75) | 3 (2 – 4) days | 3 (2 – 4) days | 4 (3 – 5) days | 4 (3 – 6) days | 4 (2 – 5) days | 3 (2 – 4) days |
| Long duration | 36.1% (n = 283) | 31.0% (n = 97) | 50.8% (n = 31) | 60.7% (n = 17) | 50.0% (n = 15) | 36.0% (n = 119) |

**Fig 3. Duration of health itinerary and timing of healthcare seeking sequence of children under-five admitted to Kisantu hospital with severe febrile illness.** Left panel: histogram displaying the duration of health itinerary of all enrolled children (n = 784), right panel: boxplot representation to follow the timing of first (formal) healthcare seeking, referral and admission of children admitted within 10 days of fever onset (n = 771). Abbreviations: BSI: bloodstream infection; NTS: non-typhoidal *Salmonella*; *Pf*: *Plasmodium falciparum*.

bloodstream infections (52.9%, 63/119) than in children without confirmed bloodstream infection (33.7%, 218/647; p<0.001; Fig 3). Health itinerary duration did not vary seasonally (median and P25 –P75 equal in wet versus dry season, p = 0.77).

**Prehospital consultation of multiple healthcare providers.** Children visited a median of two different providers, accounting for a median number of three prehospital visits per child (P25 –P75: 2–3 visits). However, the median number of visits to a formal provider was only one (Fig 4). A minority of children never sought help from a formal provider (3.6%, 28/784) and almost all (94.5%, 741/784) children were referred to the hospital by a formal provider.

Half of the children received home care, from whom only five did not consult a healthcare provider before hospital admission. Two children did not receive home care, nor any other prehospital care. Health centres were consulted by 95.4% of children and accounted for half of the healthcare visits, while health posts were rarely consulted. Pharmacies and traditional practitioners were the most consulted informal providers and were each visited by half of the enrolled children (Fig 4). Private practitioners and private health centres were together consulted by 125 (15.9%) children.

Home care and consultation of pharmacies largely occurred at fever onset. Health posts, private health centres and private practitioners were consulted earlier than health centres and traditional practitioners (Fig 4). Half (392/777) of the children who sought prehospital care went first to pharmacies and only a third (34.1%, 265/777) of children who consulted a prehospital provider went straight to a formal provider.

**Healthcare seeking as 1st factor of delay: Delayed consultation of formal healthcare providers.** The majority (56.5%, 443/784) of caretakers reported to have consulted a first healthcare provider on the day of fever onset. The minority (19.1%, 150/784) with delayed first care seeking had a significantly longer health itinerary (Table 1). Seeking care from a formal provider was delayed in 60.5% (474/784) of children and these children had a significantly longer health itinerary (Table 1).

Children with multiple prehospital visits to healthcare providers and those who visited multiple providers had longer health itineraries (Table 1). Visiting an informal healthcare provider was also associated with a long health itinerary, particularly in the case of visits to private health centres and less in the case of traditional practitioners (Table 1). When adjusted for each other, delayed first care seeking, delayed or no formal care seeking, and the number of visits were still significantly associated with a long health itinerary (S3 Table).

**Transport as 2nd factor of delay: Mostly rapid and direct transport from referring health centre.** Referral and admission occurred on the same day for all referred children (Fig 3). Most (69.8%, 547/784) children had left from the referring health centre when coming to the hospital. Almost all remaining (24.4%, 191/784) children had left from home. Transport to the hospital occurred by moto in 66.5%, by foot in 26.8%, by car in 6.6% and by bicycle in 0.1%.

Almost all (95.2%, 746/784) children arrived at the hospital within 2 hours of transport, half of them (372/746) arrived within 1 hour (Fig 1). Distributions of transport time and mode were similar in dry and rainy months. Residence in a rural village and transport time of more than one hour to the hospital were significant risk factors for a long health itinerary (Table 1).

**Prehospital patient management as 3rd factor of delay: High proportions of antibiotics, intravenous medication, and overnight stays.** Overall, almost all children received antipyretics before hospital admission and approximately one third received antibiotics or antimalarials (Fig 5). Out of 290 children who received prehospital antibiotics, 77.6% (225/290) received Access antibiotics, 17.9% (52/290) received Watch antibiotics and 11.0% (32/290) received antibiotics that were not on the national Essential Medicines List. Watch antibiotics were most

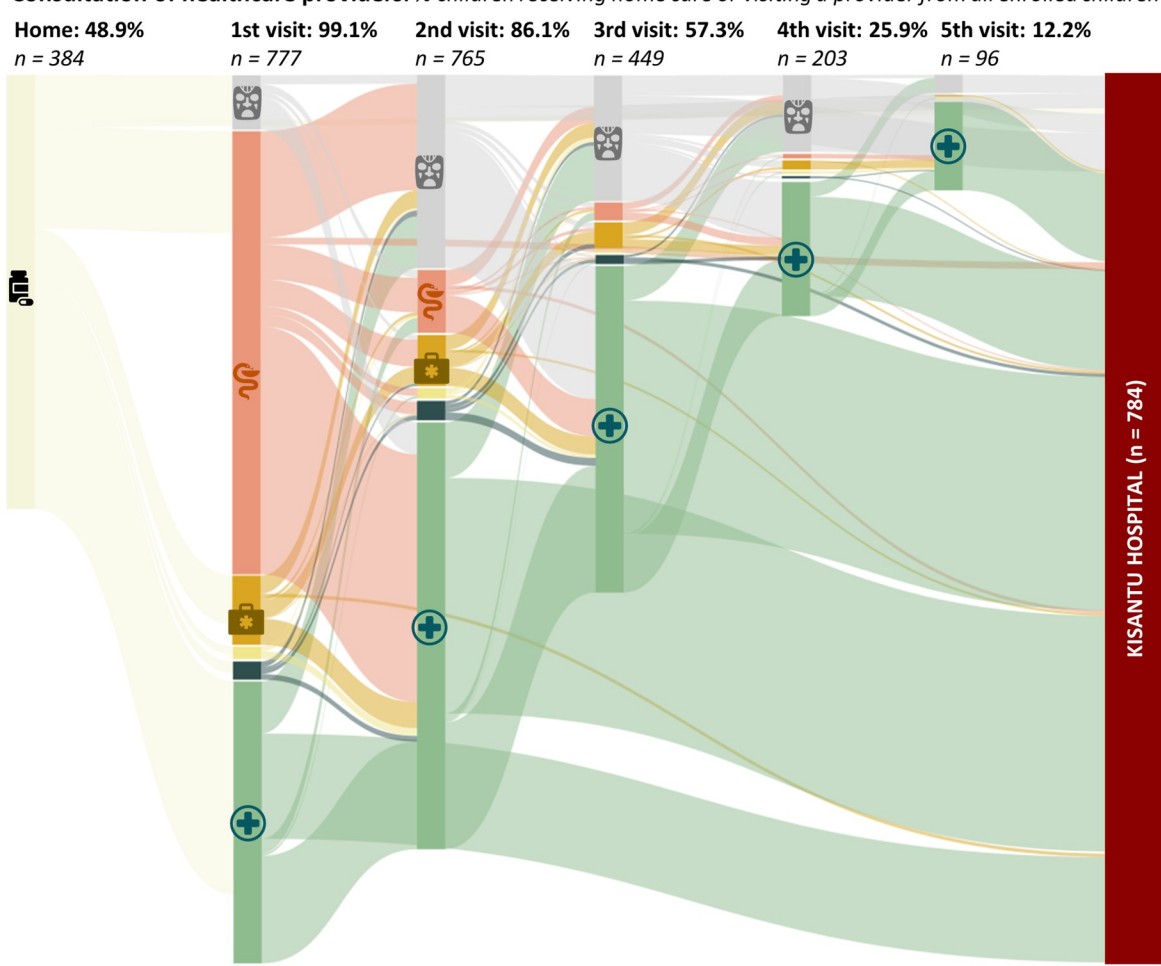

**Fig 4. Administration of home care and consultation of healthcare providers before hospital admission in children under-five admitted to Kisantu hospital with severe febrile illness (784 children accounting for 2200 healthcare visits).**

**Table 1. Unadjusted association between long health itinerary (admission >3 days after fever onset) and sociodemographic characteristics or factors from the three-delay model (healthcare seeking, transport and prehospital patient management), calculated by logistic regression.**

| n = 784 | Long health itinerary: n = 283 % (n) or median (P25 –P75) | No long health itinerary: n = 501 % (n) or median (P25 –P75) | Odds ratio (95% CI) | p-value |
|---|---|---|---|---|
| **Sociodemographic characteristics** | | | | |
| Age in months | 20 (11–31) months | 19 (9–32) months | 1.00 (0.99–1.01) | 0.54 |
| Female sex | 49.1% (n = 139) | 47.7% (n = 239) | 1.06 (0.79–1.42) | 0.70 |
| Residence in rural village | 17.3% (n = 49) | 10.8% (n = 54) | 1.73 (1.14–2.63) | **0.01** |
| Household members <5y | 49.8% (n = 141) | 53.7% (n = 269) | 0.86 (0.64–1.15) | 0.30 |
| Living with both parents (ref) | 63.6% (n = 180) | 65.8% (n = 330) | NA | |
| Living without parents | 2.5% (n = 7) | 2.8% (n = 14) | 0.92 (0.36–2.31) | 0.85 |
| Living with father | 0.7% (n = 2) | 1.4% (n = 7) | 0.52 (0.11–2.55) | 0.42 |
| Living with mother | 33.3% (n = 94) | 29.9% (n = 150) | 1.15 (0.84–1.57) | 0.39 |
| Caretaker: mother (ref) | 84.1% (n = 238) | 83.4% (n = 418) | | |
| Father | 8.8% (n = 25) | 9.4% (n = 47) | 0.93 (0.56–1.56) | 0.79 |
| Other relative | 7.1% (n = 20) | 7.2% (n = 36) | 0.98 (0.55–1.72) | 0.93 |
| Age of caretaker | 28 (23–35) years | 29 (23–36) years | 0.99 (0.98–1.01) | 0.40 |
| Health insurance [1] | 10.3% (29/282) | 11.4% (57/499) | 0.89 (0.55–1.43) | 0.63 |
| Admission in last year [2] | 46.6% (n = 132) | 42.9% (n = 215) | 1.16 (0.87–1.56) | 0.31 |
| **Healthcare seeking behaviour** | | | | |
| Home care | 49.5% (n = 140) | 48.7% (n = 244) | 1.03 (0.77–1.38) | 0.84 |
| Delayed first care seeking | 28.6% (n = 81) | 13.8% (n = 69) | 2.51 (1.75–3.61) | **<0.001** |
| Delayed/no formal care seeking | 76.3% (n = 216) | 51.5% (n = 258) | 3.07 (2.19–4.20) | **<0.001** |
| N of visits | 3 (2–4) visits | 2 (2–4) visits | 1.64 (1.44–1.86) | **<0.001** |
| N of different providers | 2 (2–3) providers | 2 (1–3) providers | 1.55 (1.33–1.81) | **<0.001** |
| Visited informal prov. | 83.7% (n = 273) | 74.9% (n = 375) | 1.73 (1.19–2.52) | **0.04** |
| Visited traditional pract. | 49.8% (n = 141) | 39.5% (n = 198) | 1.52 (1.13–2.04) | **0.005** |
| Visited pharmacy | 58.3% (n = 165) | 52.2% (n = 262) | 1.28 (0.95–1.71) | 0.10 |
| Visited private pract. | 3.2% (n = 9) | 2.8% (n = 14) | 1.14 (0.49–2.67) | 0.76 |
| Visited private health centre | 20.5% (n = 58) | 9.0% (n = 45) | 2.61 (1.71–3.98) | **<0.001** |
| Visited health post | 5.6% (n = 16) | 3.9% (n = 20) | 1.44 (0.73–2.83) | 0.29 |
| Visited health centre | 94.6% (n = 268) | 95.8% (n = 480) | 0.78 (0.40–1.54) | 0.48 |
| **Transport** | | | | |
| Mode: foot/bicycle (ref) | 28.6% (n = 81) | 28.9% (n = 145) | | |
| Moto | 65.4% (n = 185) | 67.1% (n = 336) | 0.99 (0.71–1.37) | 0.93 |
| Car | 6.0% (n = 17) | 4.0% (n = 20) | 1.52 (0.75–3.07) | 0.24 |
| Duration: < 1 hour (ref) | 40.9% (n = 116) | 51.1% (n = 256) | | |
| 1–2 hours | 52.7% (n = 149) | 44.9% (n = 225) | 1.46 (1.08–1.98) | **0.014** |
| > 2 hours | 6.4% (n = 18) | 4.0% (n = 20) | 1.99 (1.01–3.90) | **0.046** |
| **Prehospital patient management** | | | | |
| Antibiotic treatment | 54.4% (n = 154) | 29.5% (n = 148) | 2.85 (2.10–3.85) | **<0.001** |
| Antimalarial treatment | 57.2% (n = 162) | 28.5% (n = 143) | 3.35 (2.47–4.55) | **<0.001** |
| Blood transfusion | 14.5% (n = 41) | 4.0% (n = 20) | 4.07 (2.34–7.11) | **<0.001** |
| Fluid therapy | 12.4% (n = 35) | 4.6% (n = 23) | 2.93 (1.70–5.07) | **<0.001** |
| Systemic traditional care | 23.6% (n = 67) | 18.0% (n = 90) | 1.42 (0.99–2.02) | 0.06 |
| Intramuscular treatment | 71.0% (n = 169) | 52.9% (n = 265) | 1.32 (0.98–1.77) | 0.07 |
| Intravenous treatment | 38.2% (n = 91) | 12.6% (n = 63) | 3.30 (2.30–4.74) | **<0.001** |
| Intravenous antibiotics | 23.6% (n = 67) | 4.6% (n = 23) | 6.45 (3.91–10.6) | **<0.001** |
| Overnight stay | 28.3% (n = 80) | 11.0% (n = 55) | 3.20 (2.18–4.68) | **<0.001** |
| Diagnostic blood test | 84.8% (n = 240) | 67.7% (n = 339) | 2.67 (1.83–3.88) | **<0.001** |

Abbreviations: <5y: under five years old; pract.: practitioner; prov.: provider; ref: reference

[1] Three caretakers did not know if they had health insurance, these were excluded from analysis for this variable.

[2] An admission in the last year was defined as minimum one overnight stay in a health center or a hospital preceding the fever onset of the current disease episode.

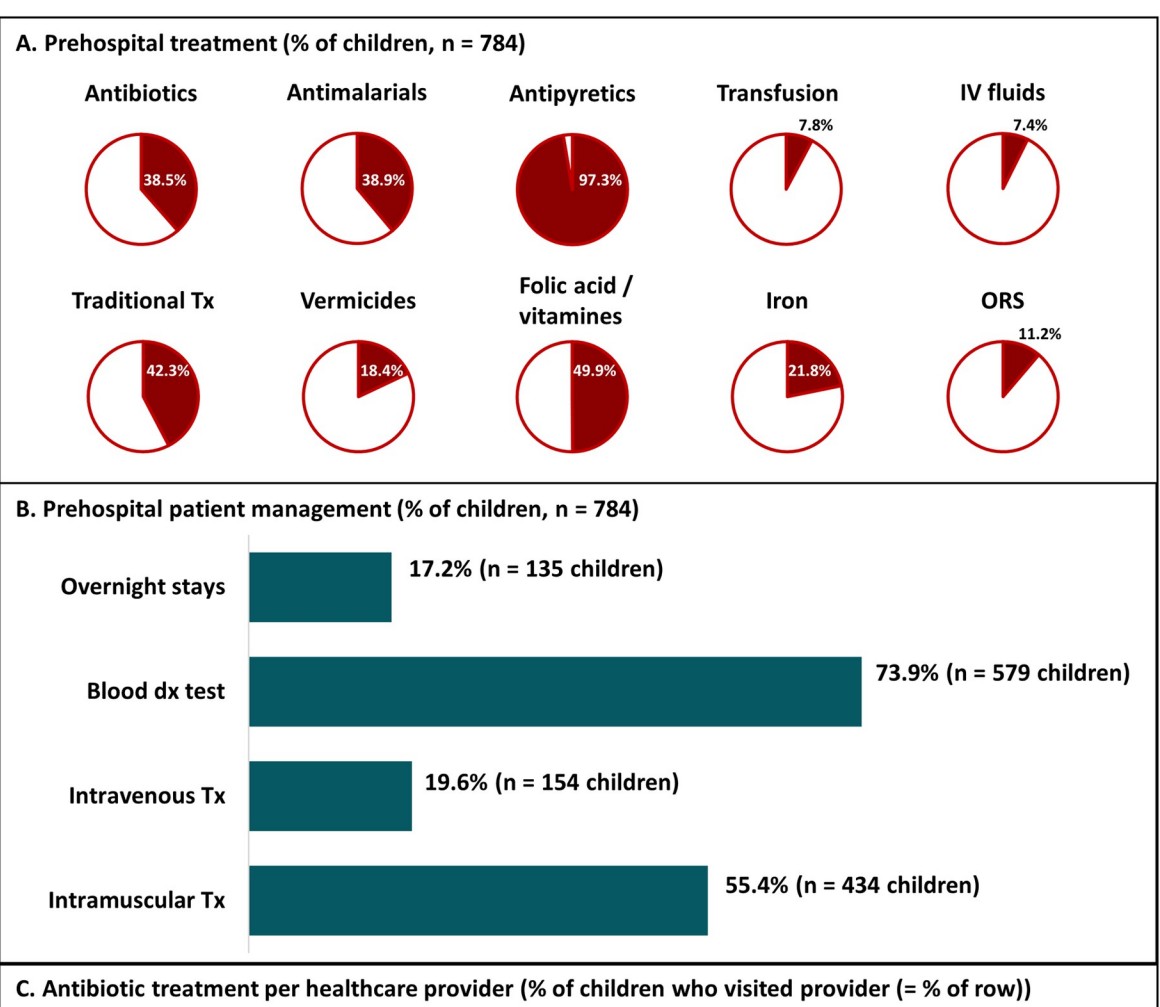

**A. Prehospital treatment (% of children, n = 784)**

| Antibiotics | Antimalarials | Antipyretics | Transfusion | IV fluids |
| 38.5% | 38.9% | 97.3% | 7.8% | 7.4% |

| Traditional Tx | Vermicides | Folic acid / vitamines | Iron | ORS |
| 42.3% | 18.4% | 49.9% | 21.8% | 11.2% |

**B. Prehospital patient management (% of children, n = 784)**

- Overnight stays: 17.2% (n = 135 children)
- Blood dx test: 73.9% (n = 579 children)
- Intravenous Tx: 19.6% (n = 154 children)
- Intramuscular Tx: 55.4% (n = 434 children)

**C. Antibiotic treatment per healthcare provider (% of children who visited provider (= % of row))**

| | All antibiotics (incl. name unknown) | Access antibiotics | Watch antibiotics | Antibiotics not on EML | Intravenous antibiotics |
|---|---|---|---|---|---|
| ✚ Health centre (n = 748) | 21.7% (n = 162) | 14.8% (n = 111) | 4.7% (n = 35) | 0.8% (n = 6) | 7.4% (n =55) |
| ✚ Health post (n = 36) | 36.1% (n = 13) | 19.4% (n = 7) | 5.6% (n = 2) | 0% | 13.9% (n = 5) |
| 🧰 Private centre (n = 103) | 59.2% (n = 61) | 34.0% (n = 35) | 11.7% (n = 12) | 3.9% (n = 4) | 27.2% (n = 28) |
| 🧰 Private pract. (n = 23) | 39.1% (n = 9) | 26.1% (n = 6) | 4.3% (n = 1) | 4.3% (n = 1) | 4.3% (n = 1) |
| ⚕ Pharmacy (n = 427) | 15.7% (n = 67) | 14.5% (n = 62) | 0.5% (n = 2) | 0% | 0% |

**Fig 5. Overview of prehospital patient management of children under-five admitted to Kisantu hospital with severe febrile illness (n = 784).** *Panel A*: Proportion of enrolled children who received at least once the respective class of prehospital care. *Panel B*: Proportion of enrolled children who had at least one overnight stay, diagnostic (dx) blood test or intravenous/intramuscular treatment (Tx). *Panel C*: Proportion of children who visited the respective provider and received antibiotics, classified according to the national Essential Medicines List (EML) if the name of the antibiotic was known by the caretaker.

frequently used in private health centres (Fig 5). We refer to S4 Table for a list of all prehospital antibiotics.

Children who received medication as home care (n = 384) were mainly given antipyretics (96%, 370/384) and rarely antibiotics (4.9%, 19/384) or antimalarials (2.6%, 10/384). Pharmacies mostly sold antipyretics (85.0%, 363/427), but also vended antibiotics and antimalarials to 15.7% (67/427) for 10.5% (45/427) of children visiting them.

One out of five children (154/784) received intravenous medication before hospital admission. This proportion increased up to nearly a third (11/36) of children visiting health posts and half (51/103) of children visiting private health centres, compared to 12.4% (93/748) of children visiting a registered health centre. Intravenous products were mainly antibiotics (57.8%, 89/154 children), blood transfusion (39.6%, 61/154 children), fluid therapy (37.7%, 58/154 children) and antimalarials (28.6%, 44/154 children). Having received intravenous medication was a significant risk factor for bloodstream infection (OR: 1.99, p = 0.003), particularly with ESKAPE-E pathogens (OR: 3.03, p = 0.006). Intramuscular treatment occurred in 55.4% (n = 434) children and was mostly confined to antipyretics (92.9%, 403/434). Traditional care was often external but involved oral administration in 47.3% (157/332) of children receiving traditional care and rectal administration in four children.

Overnight stays occurred in one out of six children (Fig 5) and were more frequent in visits to private health centres (34%, 35/103) and health posts (25%, 9/36) compared to health centre visits (12%, 91/748). During a visit to the respective provider, blood tests were performed in 91% (94/103) of children visiting a private health centre, in 75% (27/36) of children visiting a health post, and in 67% (501/748) of children visiting a health centre. No blood tests were performed by pharmacies, private practitioners, or traditional practitioners. Most (57.8%, 428/741) children did not receive treatment at the health centre during the visit of their referral to the hospital.

Prehospital antibiotics, antimalarials, blood transfusion and fluid therapy were associated with a long itinerary, as did intravenous treatment in general and intravenous antibiotics (Table 1). Children with prehospital overnight stays and children who received prehospital diagnostic blood tests had also more frequently a long health itinerary (Table 1). When adjusted for each other, prehospital antibiotics, intravenous treatment, overnight stays, and diagnostics blood tests remained significantly associated with a long health itinerary (S3 Table).

## Health-itinerary related in-hospital survival

**A long health itinerary was associated with in-hospital death.** Twenty-one children were lost-to-follow-up due to evasion from the hospital (n = 16) or referral to a specialized service (nephrology: n = 4; surgery: n = 1). Overall case fatality was 7.5% (57/763) and a large difference in case fatality was observed between children with bloodstream infection (22.8%) vs. severe *Pf* malaria (2.6%) (Fig 6). Two thirds of children dying in the hospital died during the first three days of their hospital admission (Fig 6). Furthermore, eight children died on the way to the hospital and 43 children died at the hospital before study enrolment (lag time between admission and enrolment was maximum 15 hours on weekdays, maximum 20 hours on weekend days). From the latter, bloodstream infections were confirmed in 20 children, including 16 NTS bloodstream infections.

Overall, a long health itinerary was a significant risk factor for in-hospital death (OR = 2.1, p = 0.007; Fig 6). All six children with NTS bloodstream infection who died had a long health itinerary, while a long health itinerary was less frequent in children who died with severe *Pf* malaria (50%, 4/8) or with NTS–*Pf* co-infection (61.5%, 8/13). There was a gradual increase in

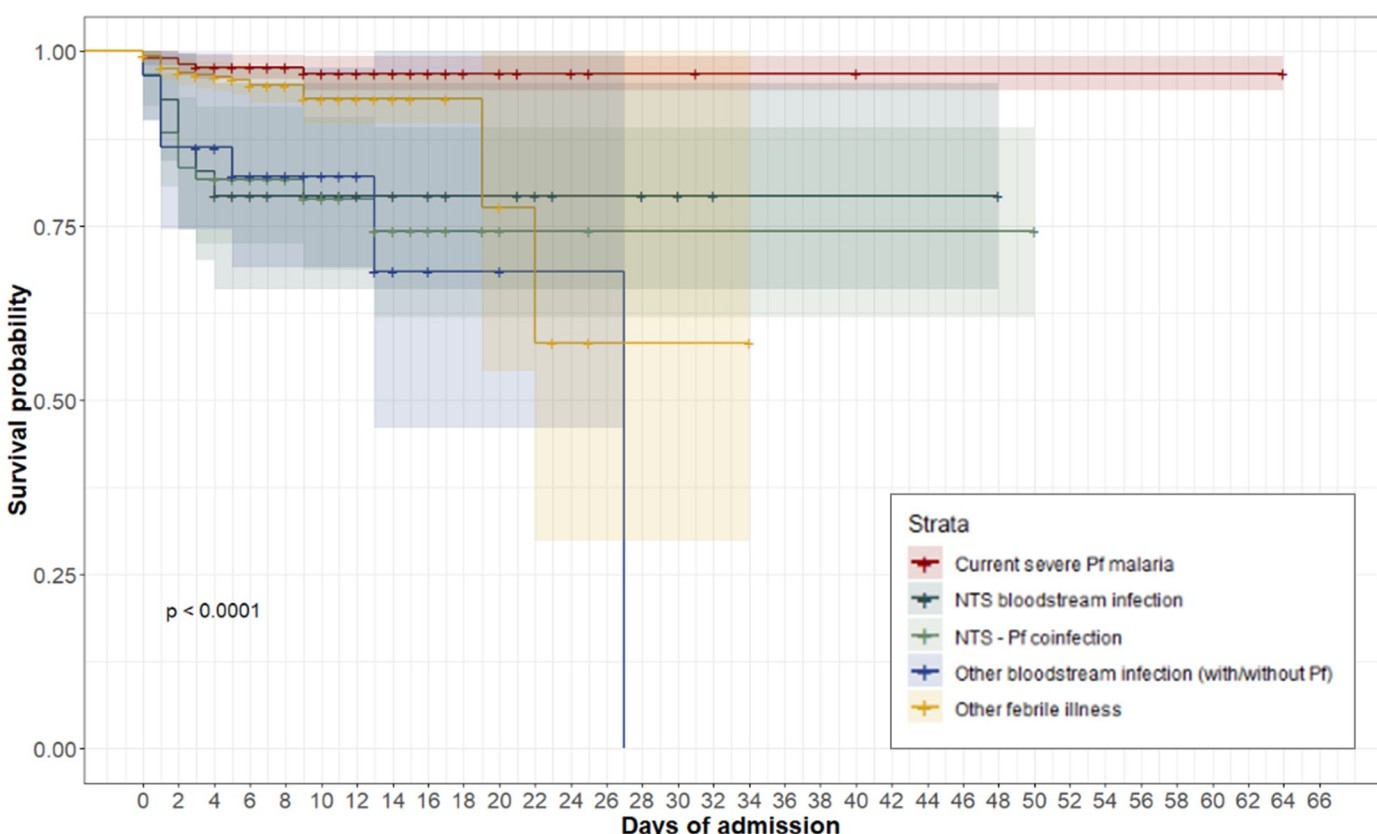

| | Overall<br><br>n = 763 | Severe<br>*Pf* malaria<br>n = 309 | NTS<br>BSI<br>n = 28 | NTS – *Pf*<br>co-infection<br>n = 58 | Other BSI<br>(± *Pf* malaria)<br>n = 28 | Other<br>febrile illness<br>n = 320 |
|---|---|---|---|---|---|---|
| **In-hospital death**<br>**% of enrolled children** | **7.5% (57/763)** | **2.6% (8/309)** | **21.4% (6/28)** | **22.4% (13/58)** | **25.0% (7/28)** | **5.9% (19/320)** |
| **Number of deaths on day 2**<br>*cumulative % of all deaths* | n = 11<br>*66.7% (38/57)* | n = 3<br>*75.0% (6/8)* | n = 2<br>*66.7% (4/6)* | n = 3<br>*66.7% (10/13)* | n = 0<br>*57.1% (4/7)* | n = 2<br>*52.6% (10/19)* |
| **Death on day 1**<br>*cumulative % of all deaths* | n = 16<br>*47.4% (27/57)* | n = 0<br>*37.5% (3/8)* | n = 1<br>*33.3% (2/6)* | n = 5<br>*53.8% (7/13)* | n = 3<br>*57.1% (4/7)* | n = 6<br>*42.1% (8/19)* |
| **Death on day of admission**<br>*% of all deaths* | n = 11<br>*19.2% (11/57)* | n = 3<br>*37.5% (3/8)* | n = 1<br>*16.7% (1/28)* | n = 2<br>*15.4% (2/13)* | n = 1<br>*14.3% (1/7)* | n = 2<br>*10.5% (2/19)* |
| **Association between**<br>**long health itinerary & death**<br>**OR (95% CI), p-value** | OR = 2.10<br>(1.22 – 3.62)<br>p = 0.007 | OR = 2.297<br>(0.56 – 9.28)<br>p = 0.25 | All deaths: long<br>health itinerary<br>(OR = NA) | OR = 1.67<br>(0.47 – 5.90)<br>p = 0.42 | OR = 1.45<br>(0.26 – 8.23)<br>p = 0.66 | OR = 1.36<br>(0.53 – 3.48)<br>p = 0.52 |
| **Long health itinerary**<br>**% of deaths** | **52.6% (30/57)** | **50.0% (4/8)** | **100% (6/6)** | **61.5% (8/13)** | **57.1% (4/28)** | **42.1% (8/19)** |
| **Long health itinerary**<br>**% of survivors** | 34.6%<br>(244/706) | 30.6%<br>(92/301) | 50.0%<br>(11/22) | 48.9%<br>(22/45) | 47.6%<br>(10/21) | 34.9%<br>(105/301) |

**Fig 6. In-hospital survival and association with long health itinerary of children under-five admitted with suspicion of a bloodstream infection according to diagnostic strata.** Twenty-one children were lost to follow-up due to hospital evasion or in-hospital referral: they were enrolled in the Kaplan-Meier curve but excluded from analysis in the table. In addition, 20 children with missing microscopy or blood culture data were excluded from the stratified analysis. Odds ratio's and p-values were calculated with logistic regression. Abbreviations: BSI: bloodstream infection; NTS: non-typhoidal *Salmonella; Pf: Plasmodium falciparum.*

**Table 2. Association between in-hospital death and sociodemographic characteristics or factors that are significantly associated with a long health itinerary (see Table 1), calculated by logistic regression.** First the unadjusted association was calculated. Next, significant unadjusted associations were adjusted for long health itinerary with multivariable logistic regression.

| Total children: n = 763 | In-hospital death: n = 57 | In-hospital survival: n = 706 | Unadjusted association | | Adjusted for long health itinerary | |
|---|---|---|---|---|---|---|
| | % (n) or median (P25 –P75) | % (n) or median (P25 –P75) | Odds ratio (95% CI) | p-value | Odds ratio (95% CI) | p-value |
| **Sociodemographic characteristics** | | | | | | |
| Age in months | 11 (7–23) months | 20 (10–33) months | 0.96 (0.93–0.98) | **<0.001** | 0.95 (0.93–0.98) | **<0.001** |
| Female sex | 51.9% (n = 29) | 48.4% (n = 342) | 1.10 (0.64–1.89) | 0.72 | | |
| Residence in rural village | 24.6% (n = 14) | 12.2% (n = 86) | 2.35 (1.23–4.47) | **0.009** | 1.98 (1.14–3.43) | **0.01** |
| **Healthcare seeking behaviour** | | | | | | |
| Delayed first care seeking | 17.5% (n = 10) | 19.1% (n = 135) | 0.90 (0.44–1.83) | 0.77 | | |
| Delayed/no formal care seeking | 61.4% (n = 35) | 60.0% (n = 423) | 1.06 (0.61–1.85) | 0.83 | | |
| N of visits | 3 (2–4) visits | 3 (2–4) visits | 1.32 (1.06–1.64) | **0.01** | 1.23 (0.98–1.55) | 0.08 |
| N of different providers | 3 (2–3) providers | 2 (2–3) providers | 1.53 (1.17–2.00) | **0.002** | 1.43 (1.09–1.88) | **0.01** |
| Visited informal prov. | 84.2% (n = 48) | 77.6% (n = 548) | 1.54 (0.74–3.20) | 0.25 | | |
| Visited traditional pract. | 59.6% (n = 34) | 42.1% (n = 297) | 2.04 (1.17–3.53) | **0.01** | 1.91 (1.10–3.32) | **0.02** |
| Visited private health centre | 24.6% (n = 14) | 12.0% (n = 85) | 2.38 (1.25–4.53) | **0.008** | 2.06 (1.07–3.99) | **0.03** |
| **Transport** | | | | | | |
| Duration: < 1 hour (ref) | 40.4% (n = 23) | 48.0% (n = 339) | | | | |
| 1–2 hours | 54.4% (n = 31) | 47.4% (n = 334) | 1.37 (0.78–2.40) | 0.27 | | |
| > 2 hours | 5.3% (n = 3) | 4.8% (n = 33) | 1.34 (0.38–4.70) | 0.65 | | |
| **Prehospital patient management** | | | | | | |
| Antibiotic treatment | 50.8% (n = 29) | 37.8% (n = 267) | 1.70 (0.99–2.93) | 0.05 | | |
| Antimalarial treatment | 49.1% (n = 28) | 37.8% (n = 267) | 1.60 (0.92–2.73) | 0.09 | | |
| Blood transfusion | 21.1% (n = 12) | 6.4% (n = 45) | 3.92 (194–7.93) | **<0.001** | 3.27 (1.58–6.76) | **0.001** |
| Fluid therapy | 19.3% (n = 11) | 6.4% (n = 45) | 3.51 (1.70–7.24) | **<0.001** | 3.03 (1.45–6.34) | **0.003** |
| Intravenous treatment | 38.6% (n = 22) | 18.1% (n = 128) | 2.84 (1.61–5.00) | **<0.001** | 2.44 (1.36–4.40) | **0.003** |
| Intravenous antibiotics | 22.8% (n = 13) | 10.5% (n = 74) | 2.52 (1.30–4.90) | **0.006** | 1.98 (0.98–4.00) | 0.06 |
| Overnight stay | 38.6% (n = 22) | 15.3% (n = 108) | 3.48 (1.97–6.16) | **<0.001** | 3.04 (1.68–5.48) | **<0.001** |
| Diagnostic blood test | 79.0% (n = 45) | 73.2% (n = 517) | 1.37 (0.71–2.65) | 0.35 | | |

Abbreviations: pract.: practitioner, prov.: provider; ref: reference

case fatality according to the length of the health itinerary, up to a case fatality rate of 13.1% for children with a health itinerary of more than 5 days. Unexpectedly, this increase in case fatality was not concentrated in the first three days of hospital admission (S1 Fig).

**Informal healthcare seeking, prehospital intravenous treatment & overnight stays were associated with in-hospital death.** In addition to their association with long health itinerary, younger age and residence in a rural village were associated to in-hospital death (Table 2), even when adjusted for the association between long health itinerary and in-hospital death (Table 2) and for each other (S5 Table).

More visits and more different providers were risk factors for in-hospital death (Table 2). In-hospital death was associated with having visited a traditional practitioner (Table 2), private health centre (Table 2) or health post (unadjusted OR = 5.56, 95% CI: 2.53–12.2, p <0.001), irrespective of the duration of health itinerary. However, when regressed together, only the association between visiting a health post and in-hospital death remained significant (S5 Table).

Prehospital blood transfusion, fluid therapy, intravenous treatment, and intravenous antibiotics were also associated with in-hospital death. Although not associated with a long health

itinerary, having received systemic traditional care was significantly associated with in-hospital death (unadjusted OR = 1.93, 95% CI: 1.07–3.49, p = 0.03). Finally, prehospital overnight stay was strongly associated with in-hospital death, even when regressed with all other prehospital patient management factors associated with long health itinerary and in-hospital death (S5 Table).

## Discussion

### Main findings

This prospective hospital-based study demonstrated that a third of children under-five admitted with severe febrile illness arrived at the hospital >3 days after fever onset. In children with confirmed bloodstream infection, this long health itinerary was present in >50% of children. A long health itinerary was associated with delayed care seeking, consultation of multiple and informal healthcare providers, living in a rural village and transport times of >1 hour to reach the hospital. A long health itinerary coincided with high proportions of prehospital antibiotics, intravenous therapy (including antibiotics, fluids, and blood transfusions) and overnight stays. One in five children with a bloodstream infection died in the hospital, compared to 1 in 50 children with severe *Pf* malaria. Having a long health itinerary was associated with twice the odds of in-hospital death. Consultation of multiple healthcare providers, consultation of private health centres, traditional practitioners or health posts, intravenous therapy and overnight stays before hospital admission were associated with higher in-hospital case fatality.

### Limitations and strengths

The single-centre, hospital-based design of this study has its limitations. Even within DR Congo, healthcare utilization differs between districts [29].The flat fee system in Kisantu district facilitates healthcare utilization as demonstrated by the fact that half of the children had already been admitted in the previous 12 months and limits the share of the private prehospital sector [15]. Furthermore, COVID-19 related changes in healthcare seeking might have occurred. Nevertheless, the study revealed the impact of public health issues that are widespread in sub-Saharan Africa, *e.g.* the mixed market of healthcare providers, poor healthcare access in rural areas, inappropriate antibiotic use and unsafe injection practices [9,10,30]. By enrolling hospital-admitted children, we only studied the tip of the iceberg and missed children with severe febrile illness who never sought care, who were never referred to the hospital, who remained in the informal sector or who died before admission. Our results are therefore likely to be a too optimistic sketch of the health itinerary of children with bloodstream infections. Nevertheless, due to the embedment of blood culture surveillance in the routine patient care, we could demonstrate that 20 children who died before enrolment had a bloodstream infection. When taking these children into account, the case fatality of bloodstream infections increased from 21.8% (26/119) to 33.1% (46/139), with the largest part of increased case fatality caused by NTS. This observation showed the importance of early enrolment for accurate case fatality estimates but also points to an underestimation of case fatality, in particular for NTS bloodstream infections.

The hospital-based recruitment also had its strengths: it resulted in a high enrolment rate and allowed us to focus on the severely ill population for which timely appropriate treatment is most crucial, to differentiate severe *Pf* malaria and bloodstream infections, and to link health itinerary to in-hospital survival. Data collection by interviewing caretakers is prone to recall and social desirability bias [31], although this was mitigated by organising the interview as soon as possible after arrival and by an independent and trained study team that was not involved in patient management. Finally, we did not perform any socio-economic or

qualitative assessment, did not assess the promptness and appropriateness of treatment upon hospital arrival, and did not study disease recognition, although the latter might be less of an issue for severely ill children [32].

## Long health itinerary and high case fatality in children with bloodstream infection

As in this study, a long duration of health itinerary in a large proportion of children was observed in other health facility-based febrile illness studies in sub-Saharan Africa [11,33–36]. Furthermore, previous verbal and social autopsy studies also revealed long health itineraries in deceased children, *e.g.* >40% of non-survivors being ill >3 days before they arrived at a health facility [11,33,37].

Two thirds of deaths occurred during the first 2 days of admission and the total number of children who arrived dead at the hospital or died before enrolment (n = 51) was almost equal to the total number of deaths in enrolled children (n = 57). Although some of these children might have been saved with good resuscitation practices, earlier presentation at the hospital is essential to allow prompt appropriate treatment and to reduce case fatality [8].

Case fatality of (NTS) bloodstream infections was ten times higher than for severe *Pf* malaria and (NTS) bloodstream infections were very frequent in children who died before enrolment. Bloodstream infections are a common and frequently fatal cause of severe febrile illness in sub-Saharan African children. Children with NTS bloodstream infection had more often a long health itinerary than children with NTS–*Pf* coinfection or severe *Pf* malaria, which suggests that non-malaria severe febrile illness is poorly recognized in prehospital care. Most children indeed had prehospital blood tests which will have facilitated malaria diagnosis but cannot detect bloodstream infections. Strikingly, none of the private practitioners performed blood tests, which contrasts with WHO recommendations for parasitological testing for all febrile patients in a malaria endemic setting [7]. Because diagnosis of bloodstream infections requires microbiological laboratory facilities that are absent at primary care level, efforts to accelerate appropriate (antibiotic) treatment of bloodstream infections should focus on the recognition of danger signs by frontline healthcare workers and early referral [4,38,39].

## Delayed formal care seeking and the mixed market of formal and informal healthcare providers

While first care seeking was prompt in most children, consultation of a formal provider was delayed in more than half of the children and was associated with in-hospital death. First care seeking often entailed buying antipyretics at the pharmacy. This practice has been previously recognized and linked to better access, lower costs, and good customer care, but does not appear to prolong the health itinerary [10,35,40–42]. Multiple visits and consultation of multiple providers prolonged the health itinerary. As previously described, caretakers hop from one provider to another if symptoms do not rapidly improve [40,43,44]. This causes interrupted treatment and increases the risk of death [11,33,40].

Consultation of traditional practitioners and private health centres prolonged the health itinerary and was associated with increased mortality. This contrasts with the community's perception of better quality of care in private centres [9], although children for whom a private centre or traditional practitioner have been consulted might have been more severely ill. In sub-Saharan Africa, there is a mixed market of private, public, and traditional healthcare providers and children often receive both biomedical and traditional services during the same febrile illness episode [9,43–47], as was observed in the confounding associations of traditional practitioners, private health centres and health posts with in-hospital death (S5 Table).

The high referral rates demonstrated that the flat fee system in Kisantu hospital encouraged referral and therefore consultation of at least one formal provider [15]. Nevertheless, as was observed in free healthcare initiatives [34,48,49], also non-financial barriers (quality of care, access, acceptability, etc.) should be addressed to improve prehospital care and accelerate the health itinerary [9,10,33,39,46,50]. Furthermore, healthcare workers at Kisantu hospital informally declared having noted abuses of the referral system, such as paid referral letters. Fake names of referring centres and a lack of data of referred patients in the health files of the referring centre were also noticed.

## The complexity of transport delays

In agreement with other paediatric febrile illness studies, health itineraries were longer in children from rural villages or in children who travelled >1 hour to reach the hospital [9,51–54]. In rural villages, families are poorer, children more vulnerable due to frequent exposure to *Pf* malaria, other infections and malnutrition and healthcare services more limited. Children from the most rural and remote health areas did not figure among the enrolled patients, which might explain why rural residence predisposed to in-hospital death, but longer transport times did not. There was no impact of the rainy season on transport times and health itinerary duration. This might be explained by a decision not to go to the hospital during heavy rains due to high transport costs, safety issues or the absence of circulating moto taxis[54,55].

## Delayed appropriate treatment due to prehospital antibiotics and injectable medicines

A third of the children received prehospital antibiotics and this was associated with a long health itinerary. The proportion of antibiotic use is comparable to national Demographic Health Survey (DHS) data in ill children under-five and is an average score for a low- or middle-income country [30]. As in a healthcare exit survey in Kisantu health district, the proportion of children who received antibiotics was highest for visits to private healthcare providers [29]. The proportion of patients who received antibiotics at health centres in the current study (21.7%) is not representative for the antibiotic use in health centres in general (51.1% in the previous survey [29]), because upon their referral to the hospital more than half of children did not receive treatment at the health centre. The proportion of antibiotic use in pharmacies (15.7%) was much lower than in the previous survey (48.8%), but the latter was based on paediatric and adult records [29].

More than 1 in 10 children received intravenous antibiotics before hospital admission and this practice was associated with a long health itinerary and in-hospital death. According to the local protocol for paediatric patient management at primary healthcare level (based on WHO Integrated Management of Childhood Illness (IMCI) guidelines [56]), intravenous antibiotics should not be used in health centres [57]. Like intravenous antibiotics, fluid therapy and blood transfusion are not indicated at primary healthcare level [56–58]. They incorrectly retain patients (overnight) at the primary healthcare level where diagnostic and therapeutic facilities are insufficient to manage bloodstream infections [5,38,39]. Intravenous therapy furthermore carries an important risk of fatal healthcare-associated infections [56,57,59,60], as observed in this study. The fact that more than half of the children received intramuscular antipyretics further illustrate the widespread problem of unnecessary and unsafe injection practices in sub-Saharan Africa [61–63].

The high share of the private healthcare sector in DR Congo carries a public health risk [10,29]. In addition to the higher total antibiotic consumption in private health centres, more Watch group and intravenous antibiotics were consumed in private centres in the current

study and in a previous survey in Kisantu [36]. Caretakers often demand "strong" medication, *i.e.* injectables and antibiotics [9,61,62], which is given more frequently in private health centres. Similar observations of high-risk practices were made in health posts, which might explain the increased risk of in-hospital death after having visited a health post. Regulation, inspection and quality monitoring of the private sector and health posts in combination with community health education to reduce prescription pressure are required to turn the tide and ensure rational antibiotic use and safe drug administration.

## Conclusions

One third of children under-five with severe febrile illness had fever for more than three days before their admission to Kisantu district hospital. Having a health itinerary of more than three days doubled the odds of in-hospital death. The fact that two-thirds of in-hospital deaths occurred before day three of hospital admission increased the biological plausibility that this was a causal association. Delaying factors associated with in-hospital death were healthcare seeking from traditional, private and/or multiple providers, rural residence, prehospital intravenous therapy (antibiotics, blood transfusion & fluids), and prehospital overnight stays. Intravenous therapy and administration of antibiotics reserved for hospital use were most frequent in the private sector. Half of the children with bloodstream infection had a health itinerary of more than three days, and one in five (excluding non-enrolled in-hospital deaths) to one in three (including non-enrolled in-hospital deaths) of children with bloodstream infection died in the hospital. Our results highlight the need for improved recognition of danger signs and earlier referral by frontline healthcare workers of the formal and informal sector, particularly in children with bloodstream infections.

## Supporting information

**S1 Table. Indications for blood culture sampling: If children fulfil both criteria when they arrive at the hospital, a blood culture is routinely sampled upon admission of the child.** (DOCX)

**S2 Table. STROBE checklist.** (DOCX)

**S3 Table. Association between a long health itinerary and sociodemographic characteristics or factors from the three-delay model (healthcare seeking, transport and prehospital patient management).** To prevent multicollinearity, a selection of variables that were significantly associated with a long health itinerary from Table 1 was made and multivariable regression was performed per delay category. (DOCX)

**S4 Table. Prehospital antibiotic consumption according to the World Health Organization's Access, Watch and Reserve (AWaRe) classification as specified on the national List of Essential Medicines (EML) list in DR Congo.** (DOCX)

**S5 Table. Multivariable logistic regression to assess the association between in-hospital death and sociodemographic characteristics or factors from the three-delay model (healthcare seeking, transport and prehospital patient management).** To prevent multicollinearity, only variables that were significantly associated with in-hospital death when adjusted for long health itinerary were selected (see Table 2 & result section) and multivariable regression was

performed per delay category.
(DOCX)

**S1 Document. Pdf version of electronic case report form.**
(PDF)

**S1 Fig. In-hospital survival of children under-five admitted with severe febrile illness according to the duration of their health itinerary.** Twenty-one children were lost to follow-up due to hospital evasion or in-hospital referral: they were enrolled in the Kaplan-Meier curve but excluded from analysis in the table.
(DOCX)

## Acknowledgments

The authors are very grateful to the HIT BSI research team for their dedicated patient enrol-ment, data collection and patient follow-up. The authors also thank all clinical, laboratory and research staff from Kisantu hospital, INRB and IVI for their collaboration during their routine clinical care and microbiological surveillance activities. We would also like to thank the hospi-tal management and local health authorities; their involvement, support and insights have been crucial for this study. Finally, the authors thank the unit of Tropical Bacteriology, the Clinical Trial Unit at ITM and the ITM office in Kinshasa for their support in writing proce-dures, clinical study management and support in logistics and accountancy. We thank Inge Van Cauwenberg for her support to the project administration and management.

## Author Contributions

**Conceptualization:** Bieke Tack, Daniel Vita, Jaan Toelen, Octavie Lunguya, Jan Jacobs.

**Data curation:** Bieke Tack.

**Formal analysis:** Bieke Tack.

**Funding acquisition:** Bieke Tack, Justin Im, Hyon Jin Jeon, Florian Marks, Jaan Toelen, Octavie Lunguya, Jan Jacobs.

**Investigation:** Bieke Tack, Daniel Vita, Naomie Wasolua, Nathalie Ndengila, Natacha Herssens, Emmanuel Ntangu, Grace Kasidiko, Gaëlle Nkoji-Tunda, Marie-France Phoba.

**Methodology:** Bieke Tack, Daniel Vita, Jaan Toelen, Octavie Lunguya, Jan Jacobs.

**Project administration:** Bieke Tack, Daniel Vita, Octavie Lunguya, Jan Jacobs.

**Resources:** Bieke Tack, Daniel Vita, Gaëlle Nkoji-Tunda, Marie-France Phoba, Jaan Toelen, Octavie Lunguya, Jan Jacobs.

**Supervision:** Bieke Tack, Daniel Vita, José Nketo, Jaan Toelen, Octavie Lunguya, Jan Jacobs.

**Validation:** Bieke Tack.

**Visualization:** Bieke Tack.

**Writing – original draft:** Bieke Tack, Jan Jacobs.

**Writing – review & editing:** Bieke Tack, Daniel Vita, José Nketo, Naomie Wasolua, Nathalie Ndengila, Natacha Herssens, Emmanuel Ntangu, Grace Kasidiko, Gaëlle Nkoji-Tunda, Marie-France Phoba, Justin Im, Hyon Jin Jeon, Florian Marks, Jaan Toelen, Octavie Lunguya, Jan Jacobs.

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
