## [Decision Letter · Decision Letter 0]

9 Feb 2023

Dear Dr. Tack,

We are pleased to inform you that your manuscript 'Health itinerary related survival of children under-five with severe malaria or bloodstream infection, DR Congo' has been provisionally accepted for publication in PLOS Neglected Tropical Diseases.

Best regards,

Paul O. Mireji, PhD

Academic Editor

Ana LTO Nascimento

Section Editor

Reviewer's Responses to Questions

**Key Review Criteria Required for Acceptance?**

**Methods**

-Are the objectives of the study clearly articulated with a clear testable hypothesis stated?

-Is the study design appropriate to address the stated objectives?

-Is the population clearly described and appropriate for the hypothesis being tested?

-Is the sample size sufficient to ensure adequate power to address the hypothesis being tested?

-Were correct statistical analysis used to support conclusions?

-Are there concerns about ethical or regulatory requirements being met?

Reviewer #1: N/A

Reviewer #2: The objectives of the study are clearly articulated where the authors assessed the health itinerary of children under-five with severe febrile illness and delaying factors related to healthcare seeking. The study design is appropriate and hase adhered to the STROBE guidelines fro this observational study.

Ethical and regulatory requirements have been met.

**Results**

-Does the analysis presented match the analysis plan?

-Are the results clearly and completely presented?

-Are the figures (Tables, Images) of sufficient quality for clarity?

Reviewer #1: N/A

Reviewer #2: Care has been taken by the authors to ensure image clarity.

**Conclusions**

-Are the conclusions supported by the data presented?

-Are the limitations of analysis clearly described?

-Do the authors discuss how these data can be helpful to advance our understanding of the topic under study?

-Is public health relevance addressed?

Reviewer #1: N/A

Reviewer #2: The study presents a clear conclusion with evidence supporting health education and health system reforms for improved recognition of danger signs and earlier referral of children under-five with severe febrile illness by frontline healthcare workers.

**Editorial and Data Presentation Modifications?**

Reviewer #1: N/A I actually thought the data was presented very well and with some interesting in and unique graphics that help in comprehension.

Reviewer #2: Minor modifications have been indicated in the annotated MS.

**Summary and General Comments**

Reviewer #1: The paper was very well written and the analysis solid. I only had 2 background questions to help understand the context of the study: 1) what was the covid prevalence (or general situation) in the area given that the study occured during the pandemic; and 2) can you please provide a brief description of the community health services in the area as context for care seeking.

Also: Line 462 had a typo. it should be 'fatal risk OF fatal healthcare....

Reviewer #2: (No Response)

PLOS authors have the option to publish the peer review history of their article (what does this mean?). If published, this will include your full peer review and any attached files.

Reviewer #1: No

Reviewer #2: No

---

## [Editor Report · Acceptance letter]

1 Mar 2023

Dear Dr. Tack,

We are delighted to inform you that your manuscript, "Health itinerary related survival of children under-five with severe malaria or bloodstream infection, DR Congo," has been formally accepted for publication in PLOS Neglected Tropical Diseases.

Best regards,

Shaden Kamhawi

co-Editor-in-Chief

Paul Brindley

co-Editor-in-Chief
